# Latent Tuberculosis Infection in Haematopoietic Stem Cell Transplant Recipients: A Retrospective Italian Cohort Study in Tor Vergata University Hospital, Rome

**DOI:** 10.3390/ijerph191710693

**Published:** 2022-08-27

**Authors:** Mirko Compagno, Assunta Navarra, Laura Campogiani, Luigi Coppola, Benedetta Rossi, Marco Iannetta, Vincenzo Malagnino, Saverio G. Parisi, Benedetta Mariotti, Raffaella Cerretti, William Arcese, Delia Goletti, Massimo Andreoni, Loredana Sarmati

**Affiliations:** 1Clinical Infectious Diseases Unit, Tor Vergata Hospital, 00133 Rome, Italy; 2Clinical Epidemiology Unit, National Institute for Infectious Diseases Lazzaro Spallanzani-IRCCS, 00161 Rome, Italy; 3Clinical Infectious Diseases, Department of System Medicine, Tor Vergata University, 00133 Rome, Italy; 4Department of Molecular Medicine, University of Padova, Via Gabelli, 63, 35100 Padova, Italy; 5Department of Biomedicine and Prevention, University Tor Vergata of Roma, 00133 Rome, Italy; 6Rome Transplant Network, Tor Vergata University Hospital, 00133 Rome, Italy; 7Translational Research Unit, National Institute for Infectious Diseases Lazzaro Spallanzani-IRCCS, 00161 Rome, Italy

**Keywords:** latent TB infection, LTBI, haematopoietic stem cell transplantation, HSCT, interferon (IFN)-γ release assay, IGRA, tuberculosis

## Abstract

The results of tuberculosis (TB) screening and reactivation in a cohort of 323 adult patients undergoing haematopoietic stem cell transplantation (HSCT) from 2015 to 2019 at the University Hospital of Tor Vergata, Rome, Italy, were reported. A total of 260 patients, 59 (18.3%) autologous and 264 (81.7%) allogeneic transplants, underwent Interferon Release (IFN)-γ (IGRA) test screening: 228 (87.7%) were negative, 11 (4.2%) indeterminate and 21 (8.1%) positive. Most of the IGRA-positive patients were of Italian origin (95.2%) and significantly older than the IGRA-negative (*p* < 0.001); 22 (8.5%) patients underwent a second IGRA during the first year after transplantation, and 1 tested positive for IGRA. Significantly lower monocyte (*p* = 0.044) and lymphocyte counts (*p* = 0.009) were detected in IGRA negative and IGRA indeterminate patients, respectively. All latent TB patients underwent isoniazid prophylaxis, and none of them progressed to active TB over a median follow-up period of 63.4 months. A significant decline in TB screening practices was shown from 2015 to 2019, and approximately 19% of patients were not screened. In conclusion, 8.1% of our HSCT population had LTBI, all received INH treatment, and no reactivation of TB was observed during the follow-up period. In addition, 19% escaped screening and 8% of these came from countries with a medium TB burden, therefore at higher risk of possible development of TB.

## 1. Introduction

Tuberculosis (TB) is still a major global public health problem. Based on the 2021 World Health Organization’s (WHO) Global Tuberculosis Report [1], approximately a quarter of the world’s population, equivalent to around 2 billion people, is infected with *Mycobacterium tuberculosis*, and 5–10% of them are expected to progress to active TB later in life [2].

The likelihood of developing active disease is much higher in frail populations, such as those with human immunodeficiency virus infection (HIV), malnutrition, cancer and haematopoietic or solid organ transplantation. Patients undergoing haematopoietic stem cell transplantation (HSCT) have severe and prolonged immunodeficiency due to underlying haematological malignancy, chemo- and immunosuppressive therapy, and graft-versus-host disease, and are generally considered to be at higher risk for active TB or reactivation of latent TB infection (LTBI).

The incidence of active TB in HSCT patients is variable and is highly correlated with the epidemiology of the disease in the country of origin (low, medium or high TB burden); overall, it has been reported to be 10 to 40 times higher than the general population, with most cases occurring within 100 days after the transplant [3].

Several studies from countries with a medium/high TB burden [4,5,6,7] have reported an increased incidence of active TB and LTBI in HSCT recipients, and in its latest guidelines on indications for TB screening, the WHO included HSCT patients among the high-risk TB population who would benefit from testing and treatment for LTBI [8].

Very little data are reported from areas of low TB burden, including most Western European countries, Canada, the United States of America (USA) and Australia. In a recent retrospective study aimed at determining the incidence of LTBI and the rate of TB among 2531 HSCT recipients from the Dana–Farber Cancer Institute and the Women’s Cancer Center in Boston, MA, USA, no cases of clinically active disease were reported [9]. In this study, only 3.6% of HSCT patients tested positive for LTBI and almost all (96%) received isoniazid (INH) treatment.

The high disease and pharmacological burden of HSCT poses additional challenges, entangling decisions not only on who to screen for LTBI but also on the timing of screening (before or after HSCT), on the treatment choice, possible drug–drug interactions and the cost–benefit balance of screening in low endemic TB countries.

The lack of studies on TB reactivation in HSCTs in low-endemic countries has led European expert groups to identify only those at greatest risk of reactivation or infection, such as patients from highly endemic countries, as those who would truly benefit from LTBI screening and therapy [10]. Italy is a country with a low prevalence of TB, and the epidemiological data show a stable TB prevalence with less than 10 cases per 100,000 inhabitants per year [11]. There is no national LTBI screening protocol for high-risk patients in Italian health facilities and a consequent lack of data on TB screening and LTBI treatment of patients undergoing HSCT.

This study aimed to describe the screening practice for LTBI in a cohort of patients who underwent HSCT over a period of 5 years at the University Hospital of Tor Vergata (Rome, Italy). The incidence of LTBI was analyzed, together with the practice of prescribing preventive therapy and the incidence of TB reactivation in the first post-transplant period.

## 2. Materials and Methods

This is a retrospective observational cohort study performed at Tor Vergata University Hospital in Rome (Italy), including all adult patients (>18 years old) with a haematologic malignancy who underwent HSCT in the Haematopoietic Stem Cell Transplant Unit from 1 January 2015 to 31 December 2019. The patients’ list was obtained by an electronic database that includes all HSCT recipients treated at the haematologic unit in Tor Vergata Hospital. An ad hoc Excel database was created to record the demographic, epidemiological, clinical and laboratory data. This study was approved by the local Ethics Committee Fondazione PTV Policlinico Tor Vergata (Protocol Number 89.22) and conducted in accordance with the principles stated in the Declaration of Helsinki. The requirement for patient informed consent was waived by the Ethics Committee considering the retrospective nature of the study, in accordance with local legislation

### 2.1. Data Collection

For each patient, information on haematologic malignancy, type of HSCT and chemotherapy conditioning regimens were recorded. Data on TB screening were collected, namely, the type, date and results of testing. All patients enrolled in the study were observed for TB active infection or reactivation until 2022, allowing a range of 2–7 year follow-up periods.

### 2.2. TB Screening Methods

Screening for *Mycobacterium tuberculosis* infection was performed through interferon (IFN)-γ release assay (IGRA) testing. Throughout the study period, two different IGRA methods were used: the QuantiFERON-TB Gold In-Tube (QFT-GIT) and the most recent QuantiFERON^®^-TB Gold Plus (QFT-Plus). While the QFT-GIT assay uses only one TB tube that mainly induces a CD4+ T-cell-mediated immune (CMI) response, the QFT-Plus has an additional TB antigen 2 tube (TB2) for the CMI response of both CD8+ T and CD4+ T cells. Patients with a positive IGRA test were evaluated for active TB presence with clinical and radiological examinations (chest X-ray and chest computed tomography (CT) scan). If active TB was excluded, the patient was considered to have LTBI. Data on LTBI prophylaxis and TB reactivation after transplant were recorded. Parallel to IGRA testing, patients were sampled to evaluate peripheral white blood cell count (WBC), neutrophils, monocytes and lymphocytes.

### 2.3. Statistical Analysis

Numerical variables were expressed as median and interquartile range (IQR) and compared among independent groups by means of the Mann–Whitney test (two groups) or by Kruskal–Wallis test (three groups) followed by pairwise comparisons with Bonferroni post hoc analyses, if appropriate. Categorical variables were expressed by numbers and percentages and compared among groups by Chi-square test or Exact Fisher test. Changes in proportions over time were assessed using the Chi-square test for trend. Statistical tests were considered significant at *p*-value < 0.05, in case of multiple comparisons Bonferroni-corrected *p*-value was <0.017. Data were analyzed using Stata (StataCorp. 2021. Stata Statistical Software: Release 17. StataCorp LLC, College Station, TX, USA).

## 3. Results

A total of 323 adult HSCT recipients treated at the Haematopoietic Stem Cell Transplant Unit of Tor Vergata University Hospital in Rome (Italy) from 1 January 2015 to 31 December 2019 were retrospectively enrolled. The characteristics of the general population, overall and according to their TB screening results, are reported in Table 1.

Most of the enrolled patients were male (56%) with a median age of 52 (interquartile range [IQR] 43–60) years. A total of 295 patients (91.3%) were Italian, while 28 (8.7%) were foreign-born, of whom 15 (53.5%) were from Eastern Europe, 5 (17.9%) were from Asia, 3 (10.7%) were from Central-South America, 3 (10.7%) were from North Africa, 1 (3.6%) was from the USA and 1 (3.6%) was from the Russian Federation. The most common haematologic malignancy affecting patients before HSCT was acute myeloid leukemia (AML) (36.6%), followed by acute lymphoblastic leukemia (ALL) (14.2%), non-Hodgkin’s lymphoma (NHL) (13.9%) and multiple myeloma (MM) (12.4%). Most patients underwent allogeneic transplant (81.7%), while 18.3% had an autologous transplant. The source of transplanted cells was peripheral blood (PBSCs) in 230 (71.2%) patients. The conditioning regimen used was a combination of thiotepa, busulfan and fludarabine (TBF) for 259 (80.2%) recipients of an allogeneic transplant and a combination of fotemustine, etoposide, cytarabine and melphalan (FEAM) for 26 (8.0%), and melphalan alone for 28 (8.7%) patients undergoing an autologous transplant. In 10 (3.1%) patients, different conditioning regimens were used. During the follow-up period, 118 patients (36.5%) died due to disease relapse and/or transplant-related complications. No infection death was TB-related. The mortality rate was higher in the group of patients who received an allogenic transplant (104/264; 39.4%) than in those who received an autologous transplant (14/59; 23.7%) (*p* = 0.024).

Of the 323 enrolled patients, 63 (19.5%) were not screened for TB before HSCT, 4 of whom were foreign-born from medium TB incidence countries (3 patients from Romania and 1 patient from Albania). All the remaining 260 patients (80.5%) were screened for TB with the IGRA test, either QuantiFERON-TB Gold In-Tube or QuantiFERON^®^-TB Gold Plus. The latter showed a higher sensitivity than the previous Gold In-Tube test, thus improving LTBI screening in the HSCT population [12]. No screening with TB skin tests was performed. In 2015, most of the enrolled patients were screened for LTBI (94.8%), while in 2019, screening rates significantly decreased, with IGRA performed on 72.3% of patients (*p* = 0.001). When comparing patients screened and not screened for TB, no significant differences in demographics, underlying haematologic malignancy, source of transplant or chemotherapy conditioning regimen were found (Table 1).

Focusing on patients who underwent IGRA testing, the vast majority (98.5%; 256/260) were screened before HSCT, specifically 240 (92.3%) within 1 month before HSCT, 13 (5.0%) between 6 and 1 month before transplantation and 3 (1.1%) more than a year before HSCT. Differently, four patients (1.5%) were tested immediately after HSCT, 3 were negative and 1 was indeterminate; these results were considered as screenings at the time of HSCT (Figure 1).

Among the 260 HSCT screened patients, 228 (87.7%) had a negative IGRA result, 21 patients (8.1%) had a positive IGRA result, and 11 patients (4.2%) had an indeterminate IGRA result (Figure 2). Of the 256 patients screened before the transplant, 22 (8.5%) underwent a second IGRA test during the first year after transplant (Figure 2): 1 in the group of 21 patients with IGRA positive, 19 in the group of 228 with IGRA negative and 1 in the group of 11 with intermediate IGRA.

Of the 21 patients with a positive IGRA, 1 patient (4.7%) underwent the test again during hospitalization for pneumonia in the first year after transplant (93 days after HSCT). In that case, sputum and bronchoalveolar lavage microscopic examination and culture, together with PCR for *M. tuberculosis*, were performed and resulted negative; therefore, the diagnosis of active TB was ruled out. The patient had an LTBI diagnosis and received INH treatment. Among the 228 patients with a first negative HSCT IGRA, 19 (8.3%) underwent a second IGRA test after HSCT: the result was confirmed to be negative for 18 patients (94.7%) but positive for 1 patient (5.3%). The patient who converted from negative to positive was tested the first time (IGRA negative) before an autologous HSCT and then repeated the IGRA test (IGRA positive) more than a year after a subsequent allogeneic HSCT. After a positive IGRA result, active TB was excluded through radiologic and microbiological tests; once LTBI was diagnosed, the patient was treated with INH for a period of 6 months. Among the 11 patients with a first indeterminate HSCT IGRA, 2 patients (18%) underwent a second IGRA after transplant, and in both cases, the result was confirmed as indeterminate.

No differences in sex distribution, geographic provenance, type or source of HSCT and haematologic malignancy were seen among patients with negative, positive and indeterminate IGRA results. No differences were noted in the overall leukocyte and neutrophil counts in patients with negative, positive or indeterminate IGRA (Table 2).

Patients with a negative IGRA had significantly lower monocyte counts than the positive IGRA groups (360 cell/mmc vs. 450 cell/mmc and vs. 450 cell/mms, respectively, *p* = 0.044). Patients with indeterminate IGRA had a significantly lower lymphocyte counts compared to the positive IGRA groups (600 cell/mmc vs. 1190 cell/mmc, respectively, *p* = 0.009).

Focusing on the 21 patients with a positive IGRA test, 12 patients (57.1%) were screened with a QFT-Plus test, while 9 patients (42.9%) were screened with a QFT-GIT test. Most patients with positive results were men (66.7%) and were significantly older than those with negative result (median age of 60 [IQR 58–64] vs. 52 [IQR 42–59] years, respectively, *p* < 0.001) (Table 2). Twenty out of 21 patients (95.2%) were Italian, and 1 patient was from Romania.

All 21 patients with a positive IGRA test were radiologically and clinically evaluated for the presence of active TB and, once the diagnosis was ruled out, treated for LTBI with INH for a period of 6 months. All 21 patients with LTBI started INH treatment 1 month before HSCT and then continued it for 5 months after HSCT. No toxicity or treatment interruption was reported in the medical records. None of the patients was treated with rifampicin. In the follow-up period, none of the 21 patients with positive IGRA developed active TB.

Sixteen of 21 (76.2%) patients with positive IGRA completed a median follow-up period of 63.4 months (range 41.6–86.8 months). Five of 21 (23.8%) LTBI patients died because of underlying haematologic malignancy or infectious complications other than TB after a median follow-up period of 16.6 months (range 2–46 months).

## 4. Discussion

In the present study, we showed the screening practice for LTBI in a cohort of patients undergoing HSCT over a 5-year period in our center. Among the 323 HSCT patients, 8.1% of the 260 patients screened with IGRA scored positive. None of them had active TB and a LTBI was diagnosed. The majority of these LTBI patients were of Italian origin (95.2%) (with only 1 foreigner) and significantly older than the IGRA-negative patients. All LTBI HSCT patients received INH treatment, and no reactivation of TB was observed during the follow-up period. To our knowledge, this is the first Italian study on the prevalence of LTBI and preventive therapy in patients undergoing HSCT.

In our population, the number of patients with LTBI was more than double that reported in a recent U.S. study by Cheng et al. [9], in which only 3.6% of 2531 screened HSCT patients were reported to have LTBI. Our data are consistent with the prevalence of LTBI in the general European population, which is higher than in the U.S. The estimated prevalence of LTBI (IGRA positivity) in the U.S. is 4.8%, with a significant decline reported since 2000 [13]. In Europe, the prevalence of LTBI in the general population is estimated to be 13.7%, with a range between 9.8% and 19.8% across countries [14]. Rates of LTBI up to 10% have been reported in the general Italian population, with the highest level in the high-risk groups [15,16], in which infection was acquired earlier during life.

Prevalence of LTBI in the global population increases with age up to double among Europeans between 50 and 80 years [14]. Regarding Italy, before 1970, the year in which the WHO identified Italy as a low TB incidence country, its prevalence was higher than 10 cases per 100,000 people; therefore, Italian subjects over the age of 50 are more likely to have been exposed to M. tuberculosis and have LTBI [17], as also shown in our cohort.

A significant and progressive decline in TB screening practices in the studied population was shown from 2015 to 2019, and approximately 19% of patients were not screened for TB. The decline in screening is likely due to the perception of a low TB risk in patients undergoing HSCT, especially if of Italian origin. This is also in agreement to recent European indications that identify only those at greatest risk of TB reactivation as those who would truly benefit from LTBI screening and therapy. However, more than 6% of those lacking screening were from a medium TB burden country (Albania and Romania) and therefore at risk for LTBI. These data reinforce the recommendation recently provided by the WHO for TB screening in groups with a TB prevalence > 0.5% and in subpopulations at higher risk factors for TB, such as those coming from high TB burden countries [18,19,20].

The number of monocytes or lymphocytes in peripheral blood or the ratio of monocytes to lymphocytes (ML ratio) has been reported to be correlated with the development of active TB [21,22,23,24,25,26]. Recent studies have shown an increase in monocytes in patients with active TB compared to subjects with LTBI and healthy controls [22]. Moreover, Rakotosamimanana N et al. showed that an increased risk of developing TB was associated with elevated peripheral blood monocyte cells and a tuberculin skin test (TST) response ≥14 mm in a group of Malagasy TB household contacts of index TB cases [25].

In one study [26] exploring whether differences in monocyte and lymphocyte counts correlated with different stages of TB infection, from LTBI to active TB, La Manna et al. showed that the monocyte/lymphocyte ratio was not a clear marker of the evolution of the infection.

In our study, an increased absolute count of peripheral blood monocytes was observed in subjects with a positive IGRA result compared to those who tested negative. Likely the indeterminate score was due to the low lymphocytes count. Moreover, monocytosis has been reported as a clear marker of neutrophil count normalization in patients recovering from myelosuppressive chemotherapy [27]; therefore, it is possible that in this category of subjects, monocytosis could be an effect of previous oncological treatment. As already mentioned, none of the LTBI patients in our cohort progressed to active TB.

Further studies on a larger cohort of HSCT recipients with LTBI are advisable to better clarify the role of monocyte and lymphocyte counts in predicting the risk of progression towards active disease.

Before drawing conclusions, a series of limitations of this study need to be considered. A longer follow-up time is required to correctly evaluate possible active TB development in immunosuppressed patients. Moreover, the study population was from a single institution, and the retrospective nature of this study limits the power of the obtained results.

## 5. Conclusions

In conclusion, a non-negligible percentage (8.1%) of our HSCT population had LTBI, all received preventive therapy and none of them progressed to active TB over a median observation time 63.4 months. A trend towards a significant decline in TB screening has been observed in recent years; this is particularly worrisome when the lack of TB screening concerns patients from high TB burden countries with a higher likelihood of LTBI and need of preventive treatment.

## Figures and Tables

**Figure 1 ijerph-19-10693-f001:**
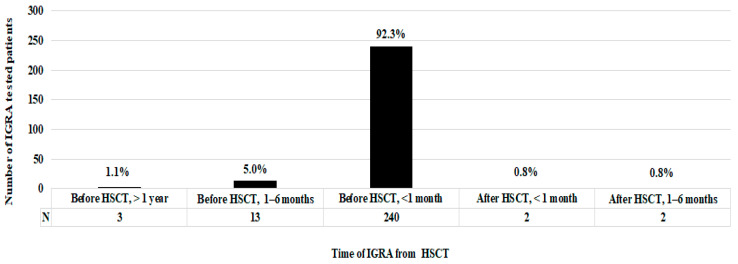
Patient distribution according to the time of IGRA from HSCT.

**Figure 2 ijerph-19-10693-f002:**
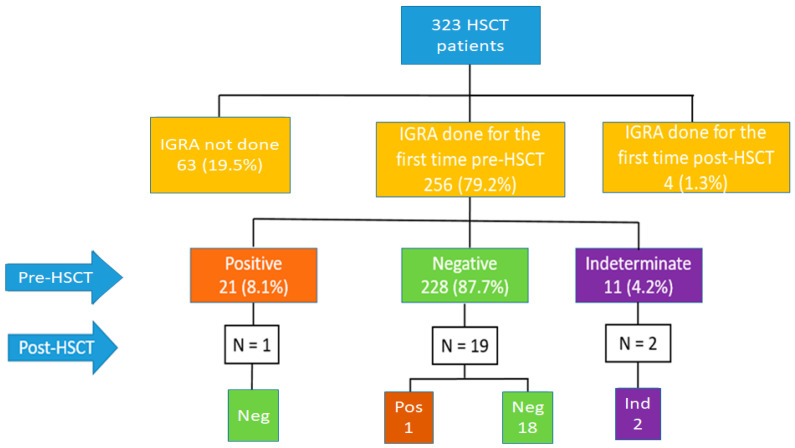
IGRA results in the overall population, both pre-HSCT and after HSCT. IGRA: Interferon gamma release assay; HSCT: haematopoietic stem cell transplant.

**Table 1 ijerph-19-10693-t001:** Characteristics of the study population, overall and stratified according to IGRA testing.

Characteristics		Total	IGRA Not Done	IGRA Done	*p* Value
		323	63 (19.5)	260 (80.5)	
		*n* (%)	*n* (%)	*n* (%)	
**Sex**	Female	142 (44.0)	26 (41.3)	116 (44.6)	0.631
	Male	181 (56.0)	37 (58.7)	144 (55.4)	
**Foreign born**	No	295 (91.3)	59 (93.7)	236 (90.8)	0.466
	Yes	28 (8.7)	4 (6.3)	24 (9.2)	
**Age, median, (IQR)**		52 (43–60)	51 (41–59)	53 (44–60)	0.509
**Year of transplant**	2015	58 (18.0)	3 (4.8)	55 (21.1)	0.001
	2016	66 (20.4)	11 (17.5)	55 (21.1)	
	2017	69 (21.4)	14 (22.2)	55 (21.1)	
	2018	65 (20.1)	16 (25.4)	49 (18.9)	
	2019	65 (20.1)	19 (30.2)	46 (17.8)	
**Type of transplant**	Autologous	59 (18.3)	8 (12.7)	51 (19.6)	0.202
	Allogeneic	264 (81.7)	55 (87.3)	209 (80.4)	
**Type of allogenic transplant**	HLA identical sibling	65 (24.6)	11 (20.0)	54 (25.8)	0.391
	HLA matched other relative	1 (0.4)	0 (0)	1 (0.5)	
	HLA mismatched relative	43 (16.3)	13 (23.6)	30 (14.3)	
	SYNGeneic	3 (1.1)	1 (1.8)	2 (1.0)	
	Unrelated	152 (57.6)	30 (54.6)	122 (58.4)	
**Source of transplant**	Umbilical cord	2 (0.6)	0 (0)	2 (0.8)	0.470
	Bone marrow	90 (27.9)	22 (34.9)	68 (26.1)	
	Bone marrow +PBSC	1 (0.3)	0 (0)	1 (0.4)	
	PBSC	230 (71.2)	41 (65.1)	189 (72.7)	
**Conditioning** **Regimen**	FEAM	26 (8.0)	5 (7.9)	21 (8.1)	0.619
	Melphalan	28 (8.7)	3 (4.8)	25 (9.6)	
	TBF	259 (80.2)	54 (85.7)	205 (78.8)	
	Other regimen	10 (3.1)	1 (1.6)	9 (3.5)	
**Diagnosis**	SAA	2 (0.6)	1 (1.6)	1 (0.4)	0.698
	BMF	1 (0.3)	0 (0)	1 (0.4)	
	CLL	3 (0.9)	0 (0)	3 (1.2)	
	AL second	10 (3.1)	3 (4.8)	7 (2.7)	
	HL	15 (4.6)	4 (6.3)	11 (4.2)	
	ALL	46 (14.2)	10 (15.9)	36 (13.8)	
	AML	118 (36.6)	24 (38.1)	94 (36.1)	
	CML	4 (1.3)	1 (1.6)	3 (1.2)	
	NHL	45 (13.9)	7 (11.1)	38 (14.6)	
	MDS	29 (9.0)	7 (11.1)	22 (8.5)	
	MM	40 (12.4)	4 (6.3)	36 (13.8)	
	MPS	10 (3.1)	2 (3.2)	8 (3.1)	

IGRA: Interferon gamma release assay; IQR: interquartile range; TBF: thiothepa, busulfan, fludarabine; FEAM: fotemustine, etoposide, cytarabine and melphalan; SAA: severe aplastic anemia; BMF: bone marrow failure; CLL chronic lymphatic leukemia; AL: acute leukemia; HL: Hodgkin’s lymphoma; ALL: acute lymphoblastic leukemia; AML: acute myeloid leukemia; CML: chronic myeloid leukemia; NHL: non-Hodgkin’s lymphoma; MDS: myelodysplastic syndrome; MM: multiple myeloma; MPS: myeloproliferative syndromes.

**Table 2 ijerph-19-10693-t002:** Characteristics of the IGRA-tested patients, overall and stratified by IGRA score.

Characteristics		IGRA Results				*p* Value
		Total	Negative	Positive	Indeterminate	
		260	228 (87.7)	21 (8.1)	11 (4.2)	
		*n* (%)	*n* (%)	*n* (%)	*n* (%)	
**Sex**	Female	116 (44.6)	105 (46.1)	7 (33.3)	4 (36.4)	0.496
	Male	144 (55.4)	123 (53.9)	14 (66.7)	7 (63.3)	
**Foreign born**	No	236 (90.8)	206 (90.4)	20 (95.2)	10 (90.9)	0.883
	Yes	24 (9.2)	22 (9.6)	1 (4.8)	1 (9.1)	
**Age, median (IQR)**		53 (44–60)	52 (42–59)	60 (58–64)	57 (49–59)	<0.001
**Year of transplant**	2015	55 (21.2)	48 (21.0)	5 (23.8)	2 (18.2)	0.998
	2016	55 (21.2)	48 (21.0)	5 (23.8)	2 (18.2)	
	2017	55 (21.2)	48 (21.0)	5 (23.8)	2 (18.2)	
	2018	49 (18.8)	42 (18.5)	5 (23.8)	2 (18.2)	
	2019	46 (17.6)	42 (18.5)	1 (4.8)	3 (27.2)	
**Type of transplant**	Autologous	51 (19.6)	44 (19.3)	7 (33.3)	0 (0)	0.074
	Allogenic	209 (80.4)	184 (80.7)	14 (66.7)	11 (100)	
**Type of allogenic transplant**	HLA identical sibling	54 (25.8)	43 (23.4)	5 (35.7)	6 (54.5)	0.301
	HLA matched other relative	1 (0.5)	1 (0.6)	0 (0)	0 (0)	
	HLA mismatched relative	30 (14.4)	27 (14.7)	1 (7.1)	2 (18.2)	
	Syngeneic	2 (0.9)	2 (1.0)	0 (0)	0 (0)	
	Unrelated	122 (58.4)	111 (60.3)	8 (57.2)	3 (27.3)	
**Source** **of transplant**	Umbilical cord blood	2 (0.8)	2 (0.9)	0 (0)	0 (0)	0.507
	Bone marrow	68 (26.1)	59 (25.9)	4 (19.1)	5 (45.5)	
	Bone marrow+PBSC	1 (0.4)	1 (0.4)	0 (0)	0 (0)	
	PBSC	189 (72.7)	166 (72.8)	17 (80.9)	6 (54.5)	
**Conditioning regimen**	TBF	205 (78.8)	180 (78.9)	14 (66.7)	11 (100)	0.139
	Melphalan	25 (9.6)	19 (8.3)	6 (28.6)	0 (0)	
	FEAM	21 (8.1)	20 (8.8)	1 (4.7)	0 (0)	
	Other regimen	9 (3.5)	9 (4.0)	0 (0)	0 (0)	
**Laboratory results**	White Blood cells/mmc, median (IQR)	3520 (2690–4850)	3445 (2690–4840)	4500 (3590–4920)	3000 (1040–4250)	0.114
	Neutrophils/mmc, median (IQR)	1905 (1185–2820)	1895 (1185–2820)	2300 (1220–2890)	1760 (550–2700)	0.821
	Lymphocytes/mmc, median (IQR)	1010 (690–1470)	1010 (690–1460)	1190 (830–2060)	600 (280–1010)	0.009
	Monocytes/mmc, median (IQR)	360 (250–490)	360 (240–470)	450 (350–530)	450 (210–530)	0.044
**Diagnosis**	SAA	1 (0.4)	1 (0.4)	0 (0)	0 (0)	
	BMF	1 (0.4)	1 (0.4)	0 (0)	0 (0)	
	CLL	3 (1.1)	2 (0.9)	1 (4.8)	0 (0)	
	LA Second	7 (2.7)	6 (2.6)	1 (4.8)	0 (0)	
	HL	11 (4.2)	11 (4.8)	0 (0)	0 (0)	
	ALL	36 (13.9)	32 (14.0)	3 (14.2)	1 (9.1)	
	AML	94 (36.1)	85 (37.3)	3 (14.2)	6 (54.5)	
	CML	3 (1.1)	2 (0.9)	1 (4.8)	0 (0)	
	NHL	38 (14.6)	34 (14.9)	2 (9.5)	2 (18.2)	
	MDS	22 (8.5)	17 (7.5)	4 (19.1)	1 (9.1)	
	MM	36 (13.9)	30 (13.2)	6 (28.6)	0 (0)	
	MPS	8 (3.1)	7 (3.1)	0 (0)	1 (9.1)	

IGRA: Interferon gamma release assay; IQR: interquartile range; PBSC: peripheral blood stem cells; TBF: thiothepa, busulfan, fludarabine; FEAM: fotemustine, etoposide, cytarabine and melphalan; SAA: severe aplastic anemia; BMF: bone marrow failure; CLL chronic lymphatic leukemia; HL: Hodgkin’s lymphoma; ALL: acute lymphoblastic leukemia; AML: acute myeloid leukemia; CML: chronic myeloid leukemia; NHL: non-Hodgkin’s lymphoma; MDS: myelodysplastic syndrome; MM: multiple myeloma; MPS: myeloproliferative syndromes.

## Data Availability

Data are deposited in an ad hoc created Excel database, available at Tor Vergata University Hospital.

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
