# Peer review of "Latent Tuberculosis Infection in Haematopoietic Stem Cell Transplant Recipients: A Retrospective Italian Cohort Study in Tor Vergata University Hospital, Rome"

_ijerph, 2022, doi:10.3390/ijerph191710693_

Round 1

Reviewer 1 Report

Dear Authors,

I very much enjoyed reading your manuscript. Your conclusion was based on good scientific evidence. 

Author Response

Manuscript ID: ijerph-1857684

Title - Latent tuberculosis infection in haematopoietic stem cell trans-plant recipients: a retrospective Italian cohort study in Tor Vergata University Hospital, Rome

Reviewer #1:
Dear Authors,

I very much enjoyed reading your manuscript. Your conclusion was based on good scientific evidence.

Answer

We thank the reviewer for appreciating our work and for the good scientific evaluation.

Reviewer 2 Report

The authors here have only represented available observations and make a case for LTBI screening pre-transplant, which considering the results in figure 2. appropriately represents their conclusion.

They have also discussed the limitations of this study and a need for a longer follow-up of a larger cohort of patients in the future, which in my opinion is sound. I, therefore, have no reservations about the publication of this study. 

Author Response

Manuscript ID: ijerph-1857684

Title - Latent tuberculosis infection in haematopoietic stem cell trans-plant recipients: a retrospective Italian cohort study in Tor Vergata University Hospital, Rome

Reviewer 2

The authors here have only represented available observations and make a case for LTBI screening pre-transplant, which considering the results in figure 2. appropriately represents their conclusion.

They have also discussed the limitations of this study and a need for a longer follow-up of a larger cohort of patients in the future, which in my opinion is sound. I, therefore, have no reservations about the publication of this study.

 Answer

We thank the reviewer for the comments and for appreciating the article without the need for revisions.

Reviewer 3 Report

Line 39 f in conjunction with Section 5. Conclusions : “In conclusion, 8.1% of our HSCT population had LTBI, 19% escaped screening and 8% of these came  from countries with a medium TB burden, therefore at higher risk of possible development of TB.”

The “conclusion” actually is not a conclusion, but more or less an epidemiological statement. In my view it appears to be more interesting to recognize that INH preventive therapy was very successful in the 8.1 % HSCCT patient who were latently MTB infected.

Lien 168 f. “All the remaining 260 patients (80.5%) were screened for TB with the IGRA test, either QuantiFERON-TB Gold In-Tube or QuantiFERON®-TB Gold  Plus. No screening with TB skin tests was performed.” Maybe it should be mentioned that the QFT plus has even a bit higher sensitivity than the preceding Gold In-tube test (Sotgiu G, et al. QuantiFERON TB Gold Plus for the diagnosis of tuberculosis: a systematic review and meta-analysis. J Infect. 2019 Nov;79(5):444-453), thus optimizing LTBI screening in the HSCT population.

Line 259: “To our knowledge, this is the first Italian study on the prevalence  of LTBI in patients undergoing HSCT.”  I suggest adding “preventive therapy” (e.g.,… study on the prevalence and preventive therapy of LTBI in patients undergoing HSCT).

Line 278 f: “The decline in screening is likely due to the perception of a low TB risk in patients 278 undergoing HSCT, especially if of Italian origin.” Is the “perception of a low TB risk” due to the observation that the risk is low after INH preventive treatment of those who were IGRA positive? Or how may be that perception may have arisen?

Author Response

Manuscript ID: ijerph-1857684

Title - Latent tuberculosis infection in haematopoietic stem cell trans-plant recipients: a retrospective Italian cohort study in Tor Vergata University Hospital, Rome

Reviewer 3

Question 1

Line 39 f in conjunction with Section 5. Conclusions : “In conclusion, 8.1% of our HSCT population had LTBI, 19% escaped screening and 8% of these came  from countries with a medium TB burden, therefore at higher risk of possible development of TB.”

The “conclusion” actually is not a conclusion, but more or less an epidemiological statement. In my view it appears to be more interesting to recognize that INH preventive therapy was very successful in the 8.1 % HSCCT patient who were latently MTB infected.

Answer 1

We thank the reviewer for the comment. The last sentence of the abstract has been modified in the article as suggested, and reported below

‘…In conclusion, 8.1% of our HSCT population had LTBI, received INH treatment, and no reactivation of TB was observed during the follow-up period. In addition, 19% escaped screening and 8% of these came from countries with a medium TB burden, therefore at higher risk of possible development of TB..’

Question  2

Line 168 f. “All the remaining 260 patients (80.5%) were screened for TB with the IGRA test, either QuantiFERON-TB Gold In-Tube or QuantiFERON®-TB Gold  Plus. No screening with TB skin tests was performed.” Maybe it should be mentioned that the QFT plus has even a bit higher sensitivity than the preceding Gold In-tube test (Sotgiu G, et al. QuantiFERON TB Gold Plus for the diagnosis of tuberculosis: a systematic review and meta-analysis. J Infect. 2019 Nov;79(5):444-453), thus optimizing LTBI screening in the HSCT population.

Answer  2

We agree with the reviewer, and we thank for the comment and for the reference. The phrase was modified as suggested and below reported. The reference has been added to the list.

.. All the remaining 260 patients (80.5%) were screened for TB with the IGRA test, either QuantiFERON-TB Gold In-Tube or QuantiFERON®-TB Gold Plus. The latter showed a higher sensitivity than the previous Gold In-tube test thus improving LTBI screening in the HSCT population.

Question 3

Line 259: “To our knowledge, this is the first Italian study on the prevalence of LTBI in patients undergoing HSCT.”  I suggest adding “preventive therapy” (e.g.,… study on the prevalence and preventive therapy of LTBI in patients undergoing HSCT).

Answer 3

We thank the reviewer; the phrase was modified as suggested

Question 4

Line 278 f: “The decline in screening is likely due to the perception of a low TB risk in patients 278 undergoing HSCT, especially if of Italian origin.” Is the “perception of a low TB risk” due to the observation that the risk is low after INH preventive treatment of those who were IGRA positive? Or how may be that perception may have arisen?

Answer

We thank the reviewer for the comment, and we agree that the sentence, as written, may give rise to poor understanding. According to the European indications of the EBMT (ref. 10 of the paper) only patients with a high risk of TB are those who benefit from screening.

At the start of the study, the policy at our center was to extend TB screening to all patients undergoing HSCT, regardless of risk and country of origin. However, it is possible that the recent international assessments on the real benefit of an expanded screening, have progressively reduced the practice of TB screening, often even with a lack of real risk assessment.

We modify the phrase in the text, accordingly. You will find it below.

A significant and progressive decline in TB screening practices in the studied population was shown from 2015 to 2019, and approximately 19% of patients were not screened for TB. The decline in screening is likely due to the perception of a low TB risk in patients undergoing HSCT, especially if of Italian origin. This also in agreement to recent European indications that identify only those at greatest risk of TB reactivation as those who would truly benefit from LTBI screening and therapy
